# Hydrophobic Effect of Soil Stabilization for a Sustainable Subgrade Soil Improvement

**DOI:** 10.3390/ma15093087

**Published:** 2022-04-24

**Authors:** Ali Muftah Abdussalam Ezreig, Mohd Ashraf Mohamad Ismail, Khaled Ibrahim Azarroug Ehwailat

**Affiliations:** School of Civil Engineering, Engineering Campus, Universiti Sains Malaysia, Nibong Tebal, Seberang Prai Selatan 14300, Pulau Pinang, Malaysia; aliezreig@student.usm.my (A.M.A.E.); khaledazrog@student.usm.my (K.I.A.E.)

**Keywords:** lateritic soil, cement stabilization, caltite, compressive strength, flexural strength, SEM, California bearing ratio

## Abstract

The chemical process of using additives to stabilize soils is to improve soil that lacks strong engineering properties. In particular, the moisture susceptibility of subgrade soil through seasonal rains is still questionable. The presence of water in the construction is the cause of deterioration and premature distress of pavements and their supporting geotechnical structures. In this work, the chemical use of hydrophobic caltite (HC) in various amounts (ranging from 3%, 5%, to 7%) and 5% of cement to enhance laterite soils is investigated. The investigation includes the evaluation of soil properties, such as, unconfined compressive strength (UCS) by curing in air and under water, flexural strength (FS), and California Bearing Ratio (CBR) soaked and unsoaked. The addition of caltite with cement increases the strength characteristics with the UCS values of 2078–2853 kPa during the early curing stages (7th day), and 4688–4876 kPa after 90 days of curing. The added caltite in the cement soil samples shows a reduction index of strength loss underwater with the UCS values of 3196, 3334, and 3751 kPa for caltite cemented soil when compared with cement soil alone. FS results suggest that the inclusion of caltite in cement means that post-peak behavior can be enhanced, reducing the brittleness and increasing the ductility. The successful reaction with soil additives occurred in the curing period of 7 days. In terms of the microstructural analysis, results show that HC with cement reduces the porosity, voids, and cracking of laterite soils. Furthermore, new polymer globules, products from the reaction, appeared on the clay particle surfaces, thereby reducing the water absorption. The addition of hydrophobic-caltite to the soil–cement mixture results in increased strength and reduced water absorption in a soil–cement mix, thus achieving a given strength value.

## 1. Introduction

Soil stabilization is a method of improving natural soil to increase its engineering qualities, such as compressive strength, shear strength, durability, and resistance to degradation in water. If water is present, then it is a significant cause of structural deterioration, premature pavement distress, and further weaknesses in various geotechnical structures that support it. Alterations to water permeability are frequently a main global factor in soil stabilization where groundwater intrusion occurs [1,2,3,4].

Lateritic soil has been used in road construction in Malaysia and in developing rural areas. The subbase of roads is constructed with lateritic soil. However, a recognized problem of lateritic soil is that it contains a high amount of clay minerals. The strength and stability could not be assured under loads, especially with the presence of water. Consequently, the bearing capacity of the poor subbase is lower than the minimum requirements of road construction. Meanwhile, rain infiltration or water could inundate the subbase and cause road damage. The three crucial features in enhancing lateritic soil minerals are as follows: (1) Enrichment of the initial material with aluminum, hydroxide, and iron oxide; (2) presence of a main kaolinite composition of the clay mineral content; (3) a high reduction in the silica content [5,6,7].

Almost every tropical and subtropical region with abundant residual lateritic soils extensively use chemically stabilized soil–cement mixtures. The mixtures can improve the engineering properties of poor lateritic soils [8,9]. For instance, cement and lime are extensively utilized in numerous roadbed treatment projects. The benefits include high levels of integrity and stability and low costs [10,11].

However, although adding cement to soil makes it more robust and stiffer, it also makes it more brittle. The situation is because every pavement structural layer encounters tensile or flexural stresses. The effects could be seen on the development of dams, barriers, or pavements of multiple layers due to the flexure of tensile stress zones in earthen features. Tensile strength must be considered a critical design feature when embedding reinforcement materials in soil [12,13]. Some crucial parameters in designing pavements are flexural strength (FS), unconfined compressive strength (UCS), and soaked California Bearing Ratio (CBR) [14,15].

However, several tests showed that cement soil is unsuitable for specific projects due to certain defects, such as low tensile strength, great brittleness, and poor deformability [16,17]. In response to the above-mentioned problems, many researchers have proposed various improvement methods based on different materials [18,19,20,21]. Few studies discovered that the use of fiber reinforcement technology to enhance mechanical properties effectively improved the soil [22,23,24,25].

Recently, different liquid chemicals, such as sodium silicate, magnesium chloride, sulfonated oil, canlite, and probase, are used to improve the compressive strength in road stabilization projects [6,26,27,28,29,30]. Those chemicals are used in unconfined compressive testing on lateritic soil samples of varying contents. The tests revealed an initial increase in compressive strength before it decreased with the increase in the sample’s material content. A necessary supplement to the examination of compressive strength is to assess the tensile (flexural) of the bound materials. The FS is governed by any failure of the cement-treated base (CTB) [31].

Some studies indicated that the strength of cement-stabilized lateritic soils is susceptible to moisture levels even after 28 or 90 days of curing. Given that the stabilized lateritic soil strength dramatically reduces after water immersion, researchers suggested that a study on soaked and unsoaked strengths should also be conducted to avoid design errors [32,33]. To the best of our knowledge, no research has been shown to date on the use of caltite in pavement geotechnics to improve the durability of cement-stabilized pavement base/subbase, which is the subject of this study. This study is the first attempt to use caltite with ordinary Portland cement (OPC), making it novel.

Several experimental studies that examine the feasibility of using local cement-treated lateritic soil for constructing roads is presented in this study. This study aimed to evaluate the mechanical behavior of hydrophobic caltite (HC) on the cement soil exposed to water ingress. A 5% of OPC and various percentages of HC were mixed with lateritic soils. The preparation involved changing the content of molding water. The soil was subjected to geotechnical and microscopic tests to determine their initial material characteristics before adding cement and caltite. Testing involved compaction, unconfined compressive strength (UCS) in unsoaked and soaked conditions, flexural strength (FS), porosity, and absorption (PA), California Bearing Ratio (CBR) test, and scanning electron microscopy (SEM). The outcomes will provide experimental evidence to evaluate the practical usage of caltite-reinforced cement soil in engineering projects.

## 2. Materials and Methods

### 2.1. Subsection

#### 2.1.1. Soils

The study area is in the northwest part of Malaysia, where lateritic soil is abundant. Excavations of lateritic soil were made 1 m below the ground surface in Permatang Kerat Telunjuk, Bandar Baharu, Kedah, Malaysia. The classification of the lateritic soil was tested following British Standards [34,35], and the results are presented in Table 1. According to the Unified Soil Classification System (USCS), the soil was classified as silty sand (MS).

#### 2.1.2. HC

Caltite, a unique double-action and a waterproofing compound, could line the pore and capillary structure of the cement paste, reversing the standard capillary suction and physically blocking the capillaries when subjected to hydrostatic pressure. The precise chemical composition of the additive is not in the public domain because as it is a commercially registered brand [36].

#### 2.1.3. OPC

In this study, the OPC conforming to BS EN 197-1:2011 [37] was used to bind the stabilized soil. The OPC was manufactured by Cement Industries Malaysia Berhad (CIMA). Care was taken to ensure that the cement did not exceed the 3 month storage period after production.

### 2.2. Method

#### 2.2.1. Mix Design

The first soil mixture with 5% cement was prepared, as recommended in previous studies [38,39]. Next, the second mixture was prepared with different percentages of caltite (5%, 10%, and 15%) before being added to the cement soil mixture. The added cement and caltite were based on the soil dry weight. In this study, the mechanical tests used were standard proctor (SP) compaction tests (BS EN 13286-2:2010) [40], UCS tests (BS EN ISO 17892-7:2018) [41], CBR (BS EN 13286-47:2012) [42], and FS tests (ASTM D1635 2019) [43]. All the experimental tests and procedures conducted in this study were based on the British Standards.

#### 2.2.2. Soil Sample Preparation

After collection, the soil samples were oven-dried at 50 °C for 24 h to substantially reduce the soil water content. After crushing and sieving, the soil was sieved with a 2 mm gap [16] to ensure size uniformity. This study utilized soil, cement, and caltite as three principal types of material. Deionized water was used as a solvent throughout the sample preparation due to its suitability for chemical tests. Subsequently, the mixture was thoroughly mixed until it achieved a uniform blend. The designation mix for this study is presented in Table 2.

#### 2.2.3. Experimental Procedure

The optimum moisture content (OMC) for the natural lateritic and the various combined designs could be determined because the samples were subjected to SP compaction tests (BS EN 13286-2:2010) [40]. When the soil sample was in its natural state and mixed with cement and caltite at 5% each (or both of these in equal 5% proportions), the maximal dry density and OMC [44], or the required water quantity was identified.

Palette knives were used to hand-mix and prepare the homogeneous mixtures ahead of the tests for UCS, FS, and PA content. Next, each sample underwent compression in steel, cylinder-shaped mold to obtain the required dry density and moisture content. Before adding materials to the next layer, the surface of each layer was scraped. This process could interlock the layers and limit the potential instances of the specimens horizontally cracking. Next, the samples were compacted with a hydraulic jack. Finally, each sample was extruded from the mold with 100 mm height and 50 mm diameter using a steel plunger (diameter = 45 mm).

Subsequently, each sample was trimmed and wrapped in a polythene cover and an aluminum sheet to retain the moisture content [45]. In the first method, each sample that underwent UCS testing was cured for 7, 28, and 90 days. Before the test, the samples were immersed in water for 24 h and air-dried for at least 2 h at room temperature after curing periods of 7, 28, and 90 days. In the second method, the samples were cured under water at 7, 28, and 90 days after a curing time of 1 day in air and wrapped in a polythene cover and an aluminum sheet to retain the moisture content. Testing was conducted on at least three specimens for each mixture to provide an index of soil improvement. Lastly, SEM analysis and energy dispersion X-ray (EDX) were conducted to observe the changes in sample morphologies and the interpretation of the reaction after the stabilization process.

## 3. Results

### 3.1. Characteristics of Strength Development Analysis

#### Influence of Caltite Content and Cement on Soil Density

Maximizing the soil density is a crucial objective of compaction in this study. The addition of a single anion or cation due to stabilizers might significantly alter the soil property compaction. Accordingly, the initial compaction effort influences the reactions of the soil stabilizers due to the direct impact on particle spacing and the subsequent crystallization process. Compaction tests were conducted on three forms of soil: pure lateritic, soil combined with 5% cement alone, and soil mixed with 5% cement and 3%, 5%, and 7% liquid stabilizer caltite.

The addition of caltite and cement to the lateritic soil, whether mixed or separately, resulted in a slight reduction in dry density and an increase in OMC for an identical compaction input (Figure 1). However, the extent of the difference was relatively minor. The dry density reduction could be explained by particle flocculation and accumulation caused by the rapid exchange of cations in the soil–stabilizer mixture [46,47].

Cation exchange causes particles to flocculate, agglomerate, and become slightly coarser, resulting in a drop in MDD. The larger and aggregated particles, which are welded together by strong interparticle connections, resist denser packing, resulting in a reduced dry unit weight. Flocculation also increased the amount of vacant space and the size of the voids, resulting in a higher water demand to fill them [46].

Furthermore, the rise in moisture level was most likely due to the soil-additive reaction’s swift and exothermic nature, which resulted in water loss. The data gathered in this study and the research outcomes by [46,48,49] demonstrated good agreement with the examined cement soil.

### 3.2. Influence of Caltite Content and Cement on UCS

A UCS test was conducted to determine the effectiveness of the chosen additive in increasing the compressive strength of the three types of soil. The soils were natural lateritic, soil mixed with cement, and cement mixed with varying proportions of caltite (Table 2). The compressive strength and processing times for the NLS, LS + HC, LS + OPC, and LS + OPC + HC mixtures unsoaked condition are plotted in Figure 2. Meanwhile, Figure 3 illustrates the outcomes of the UCS soaked condition test performed on the cement soil and stabilized combinations of lateritic soil containing optimum caltite percentage (5%) with cement (5%) after 7, 28, and 90 days of curing. The result showed that the lateritic soil strength characteristics were significantly enhanced following treatment with cement. Specifically, the addition of cement improved the lateritic soil during the curing periods. When 5% caltite was added to the mixed cement and soil, the result increased more than when 5% cement alone was added.

Meanwhile, a smaller rise in UCS was observed when 3% of caltite was added. By contrast, the UCS increased when 5% caltite was added. The differences in compressive strength observed in the soil with the 5% cement mixture and 5% caltite addition appeared to be the greatest. As previously mentioned, a rise in UCS occurred when the period of curing continued (Figure 2). However, no significant effect of the caltite solution alone was observed on the soil.

However, the most significant growth in strength occurred during the initial 7-day curing period. The combination of treated samples with 5% caltite–cement (LS + OPC + HC) reached a compressive strength of 2853 kPa after a 7-day curing period compared with the addition of 5% cement alone (2078 kPa). This pattern represented a value around eight times higher than the untreated soil strength when 5% of cement was added to the soil and eleven times when 5% of caltite was added to cement-treated soil.

This strength developed faster during the initial 7-day period than during the subsequent periods (for 28 days, an increment of 230 kPa was observed). The mixture of 5% caltite, cement, and soil achieved a UCS value of 2853 kPa during the initial 7-day curing period (an increment of 753 kPa), while a final value of 4876 kPa was attained after the 90-day curing (an increment of 188 kPa) compared with the 5% cement soil mixture (Figure 2).

This situation implies that most reactions with soil stabilizers occurred during the initial curing stages. When immersed in water for 24 h and air-dried at least 2 h at room temperature, the strength of caltite-cement soil (LS + 5% OPC + 5% HC) increased to 3000 and 4235 kPa, respectively, during the 7–28 days with a slight strength loss to 4840 kPa at 90-day curing periods. By contrast, the cement soil samples (LS + 5% OPC) with the same conditions reported an increased strength of 2606 kPa during the 7-day curing period. Meanwhile, the strength development decreased to 3429 and 4230 kPa during the curing periods of 28 and 90 days. The drop was approximately 9% with the soil and cement sample. By contrast, no decrease in strength, but rather an increase, was observed with the cement, caltite, and soil sample (Figure 3).

Conversely, the compressive strength of the samples treated with a cement alone and caltite liquid additive was reduced when the samples were cured 1 day in air and immersed in water for 28 and 90 days (curing under water), except for 7 days of curing wherein an increase in UCS until 2550 and 3196 kPa of mixtures was recorded. Meanwhile, the reduction rates in UCS at 28 and 90 days were 3054 and 3206 kPa for the cement soil samples (LS + 5% OPC) and 3334 and 3751 kPa for caltite–cement soil (LS + 5% OPC + 5% HC), respectively (Figure 4).

The reduction in UCS in curing under water caused water ingress to the cement soil samples. Specifically, cement soil absorbed water in the first 7 days, and the water caused an increase in the reaction with calcium and in strength in the short term. Thereafter, calcium was consumed early and led to calcium leaching and a decrease in strength in the long term.

The compressive strength of the samples treated with a caltite liquid additive showed more than 5% reduction. According to [27,50,51], the most likely explanation was the increment of a positive surcharge and soil particle repulsion within the resulting mix. The chemical stabilizer effect might also explain these outcomes. A small proportion began to fill the soil voids, thus acting as a cementing bond. Once the chemical stabilizer had filled each void in this manner, the increased softening state of the mixture that exceeded the requirement for chemical reaction, resulted in a strength reduction.

Furthermore, the stabilizer percentage affects clay minerals. A small caltite percentage was sufficient to produce a clay platelet coating. Nonetheless, an increase in the stabilizer percentage in the soil body acts as a lubricant as the clay particles slip across one another. This condition has reduced the soil skeleton strength. Therefore, in this study, 5% caltite can be considered as the optimal design for a soil mixture, along with the addition of 5% cement to lateritic soil for micro-characterization.

### 3.3. Influence of Caltite Content and Cement on FS

The flexural behavior of the soil was significantly altered when caltite was added to the specimen with cement. The alteration elevated the FS value while also improving the stiffness of the specimens. Figure 5 shows a comparison of the flexural loads and deflection curves of the NLS, LS + HC, LS + 5% OPC, and LS + OPC + 5% HC mixtures. The host soil (LS) exhibited a frail bending resistance, with values of 178.11, 210.36, and 277.23 kPa after the 7, 28, and 90 days of curing periods, respectively.

Moreover, the soil with the caltite mixture alone (LS + 5% HC) had a weak bending resistance, measuring only 149.86 and 168.01 kPa. The weak bending resistance was most likely due to the lack of calcium hydroxide, which is the basis of the reaction with caltite. Further explanation will be discussed in the microstructural analysis.

The LS + 5% OPC specimen demonstrated improved bending resistance at 980.30, 1334.59, and 1497.73 kPa after the 7-, 28-, and 90-day curing periods. The improvement was probably due to the formation of calcium aluminate hydrate (C–A–H) and calcium silicate hydrates (C–S–H) because the cement content was hydrated. This result is consistent with previous studies [52].

Meanwhile, the mixture specimens (LS + OPC + 5% HC) exhibited significantly earlier improved resistance to 1403.43, 1541.7, and 1637.61 kPa after 7, 28, and 90 days of curing periods, respectively (Figure 5). The trend was substantial due to the reinforcement of 5% HC, which was introduced during the periods. The 5% HC is a considerable enhancement in bending resistance and was greater than that observed in the LS + 5% OPC specimens, as shown in Figure 5.

### 3.4. Influence of Caltite Content and Cement on Porosity and Absorption

The characteristics in terms of porosity and water absorption of soil are important factors in the evaluation of the strength properties of soil. The soil permeability has a direct link with the pore size, structure, and interconnectivity. Consequently, the pore connection causes high permeability, which could result in poor strength properties and durability [53]. This test was used to determine the percentage of porosity and water absorption by the stabilized specimens by total immersion of the samples into the water after being cured for 7, 28, and 90 days. Figure 6 shows the percentage of porosity and water absorption of the stabilized specimens with cement soil and cement soil with caltite. The observation result shows that, in all systems, the percentage of porosity and water absorption decreases with an increase in curing time. Meanwhile, the percentage of porosity and water absorption of the stabilized specimens by (MC + OPC + HC) significantly decreased when compared with cement alone. These figures demonstrate that when 5% of caltite (MC + OPC + HC) was added to the cemented soil, the soil properties improved by reducing the rates of water absorption and porosity at rates ranging from 2% to 3.5% with caltite at all ages compared with the cemented soil alone mix that was (MC + OPC). Furthermore, the use of caltite of 5% of cement soil decreases the water absorption and porosity at rates greater than with increasing curing time.

### 3.5. Influence of Caltite and Cement on CBR

The CBR is a measurement system used to describe the road subgrade strength. The CBR parameter determines the thickness of the pavement and the layers it comprises [54]. Lateritic soil has CBR values that are relatively low (<15), making them unsuitable for road subgrade. The study findings revealed substantial CBR gains (Table 3). A moderate but significant increase in strength was observed in the LS + OPC and LS + OPC + HC mixtures, making the tested soil a suitable material for road construction. The findings of the cement-treated soil tests were in line with [49,55]. The use of the mixture (LS + OPC + HC) significantly influenced the CBR rise.

In the case of the unsoaked specimens after the 7-day curing period, the CBR values ranged from 109% to 115% (LS + OPC) and 114% to 120% for the LS + OPC + HC mixture. When HC with the cement was used, the soil samples showed a noticeable trend for the CBR to significantly increase over time (Figure 7). Except for the LS + HC mixture, the CBR value was significantly reduced because this HC had no positive impact in the absence of cement. The increase in the moisture content was probably because of the rapid and exothermic nature of the reaction between the soil and the additive, resulting in water loss. However, the soaked CBR samples in water for 4 days, after curing for 7 days, with LS + OPC + HC soil mixture yielded higher values than the soaked LS + OPC. The soaked CBR values for the lateritic stabilized soil are illustrated in Figure 7.

## 4. Discussion of the Effect of HC with Cement on Treated Soil

The principal active ingredient in caltite is a soluble ammonium stearate, which accounts for 30% of its weight. Caltite shows chemical and physical effects on soil properties when cement is present, and shows no effect in the absence of cement. The chemical impact takes place at the ammonium stearate stage. This situation produces calcium hydroxide in the soil–cement paste, resulting in the formation of calcium stearate and the release of ammonia gas. Some studies suggested that calcium stearate contributes to soil improvement and reduces water absorption [56].

Consequently, the insides of the capillary pores are coated with a water-repellent membrane, causing capillary suction in the soil to reverse. The caltite physical effect occurs during the hardening process. The bleed water moves the suspended microscopically small, asphalted particles (polymer globules) in caltite and collects them in the capillary voids. Figure 8 describes the mechanism of gathering polymer globules. The polymer globule mechanism was observed using SEM, as shown in Figure 9. These particles could block the capillaries if the hardened soil is subjected to the ingress of external moisture, especially when the soil meets hydrostatic pressure. This situation happened when the soil is soaked in water for 24 h after drying for at least to 2 h at room temperature before the UCS test. Subsequently, the soil strength increased more than the dry condition at 7 and 28-day curing times, as shown in Figure 3. This finding is comparable with previously reported studies in concrete technology [36,57].

### 4.1. Microstructural Analysis

The microstructural characterization can provide essential information concerning the mechanisms that underlie chemical stabilization processes. These include the element compositions, changes in mineralogy, and the in-filling of vacant pore spaces caused by cementation. Based on the outcomes of the tests used in this study, the optimal HC and cement levels for all stabilizations were ascertained to be 5%. As a result, the microstructural analysis was aimed to compare 5% cement with and without HC-stabilized lateritic.

#### 4.1.1. SEM Analysis

SEM was utilized to observe the morphologies of treated soil samples after a 28-day curing period. Figure 9a,b shows that the lateritic–cement mixture underwent changes in its microstructure that culminated in a denser structure. However, the cement-stabilized soil micrograph showed the presence of voids, porosity, and cracking, which primarily contributed to the loss of strength when underwater. These cracks act on continuous water ingress when the cement soil was immersed in water; this interpretation was in agreement with the previous studies [32,58]. Meanwhile, the cement produced a new material when treated with a caltite mixture. Hence, the cracked soil structure and porous regions were filled, as shown in the SEM images in Figure 10a,b.

As can be seen in Figure 10a,b, white polymer globules formed inside the soil during the 28-day curing period. The SEM micrograph revealed that significant alterations occurred such as formation of gel-like products in the soil microstructure. The alterations had layered the soil surface. Therefore, the increase in the sample shear strength was due to the contribution of these new formations.

The formation of polymer globules in the early stages of curing also indicated that the cement–caltite reaction was rapid. Porosity, voids, and cracking were observed in a soil mixture containing cement. In particular, the new products occupied various porous parts and micro-cracks in the soil structure.

#### 4.1.2. EDX Spectroscopy

Figure 11 and Table 4 show each EDX spectrum for the untreated lateritic soil and the soil with HC and cement additive after compressive strength tests at a curing period of 28 days. The untreated lateritic soil showed comparably high intensities of Si and Al peaks, which corresponded to the soil lateritic nature [51].

Table 4 and Figure 11a–c illustrate how the Si/Al and Si/Ca percentages increased in the caltite mixed with cement-treated soil compared with the cement-treated soil alone and the untreated lateritic soil. The reaction between the caltite and the cement caused an increase, indicating the possibility of a change in the surface particle composition. The changes resulted from the pozzolanic reaction coating or the production of a new mineral content in the soil with cement and caltite and cement alone. The hydroxyl group (OH) was present and increased in percentage due to the reaction between the caltite and cement. This situation produced new polymer globules, which accumulated in the capillary voids and prevented water from escaping or evaporating from the soil sample. Next, a pozzolanic reaction would occur if Ca continued to be available. The reaction would result in the formation of Ca hydrates because OH, Ca^2+^, AlO_6_, and SiO_4_ are present [59,60].

Additionally, the cementation due to the C–A–H and C–S–H pozzolanic reaction depends on the percentage of Si/Al. Increased pozzolans, which are rich in silica and alumina, contribute to the formation of C–A–H and C–S–H. Thus, the formed binder unquestionably contributes to the higher UCS found in the cement and caltite-modified soils compared with the non-treated soil. The process could be explained by the interlocking arrangement that occurred between soil particles [61,62].

## 5. Conclusions

Implementing hydrophobic material in cement technology is a crucial feature currently under development. Hence, the investigation into one method of utilizing HC-cement treatments to stabilize lateritic soil is vital. A significant discovery of the current study was that a combinatory action between the cement and caltite overcame stabilization problems. The following are the summarized concluding details from the laboratory experiments:The addition of caltite to soil treated for cement stabilization increases the UCS, shear strength, FS, and CBR values in the two chemical agents.Adding caltite to cement-treated soil improves its post-peak behavior, particularly during the early curing stages (the 7th day). In this case, the best soil-additive reaction occurred.Stabilizing the soil with reinforced HC resulted in a greater strength than cement alone, regardless of the curing period. This trend is most noticeable after the 7, 28, and 90 days of curing periods.After 7 and 28 days of curing, the soil treated with caltite and cement showed a significant pozzolanic reaction compared to the soil treated with cement alone or untreated lateritic soil.The interfacial interlocking observed in the soil treated with HC and cement-treated soil demonstrated effective bonds compared to that with untreated soil. However, the mixture of HC and cement reduced porosity, voids, and cracking in the soil sample compared with the cement soil sample.SEM and EDX analyses revealed that the reaction that occurs in the soil treated with caltite and cement mixture resulted in a denser soil. By contrast, the reaction that occurs in the soil alone tends to result in the binding of soil particles and the strengthening of soil.Finally, the SEM analysis proved that adding caltite to cement soil led to new globular polymer bonding compounds. These compounds essentially filled the soil cracks and porous voids. Consequently, the soil density improved with greater strength and more effective void filling.

In conclusion, adding caltite to cement and soil mixture can significantly strengthen the treated soil, resulting in suppression of strength loss in cement-treated soil when using in wet environments. Given that this material is hydrophobic, future research should focus on improving the soil in other environments.

## Figures and Tables

**Figure 1 materials-15-03087-f001:**
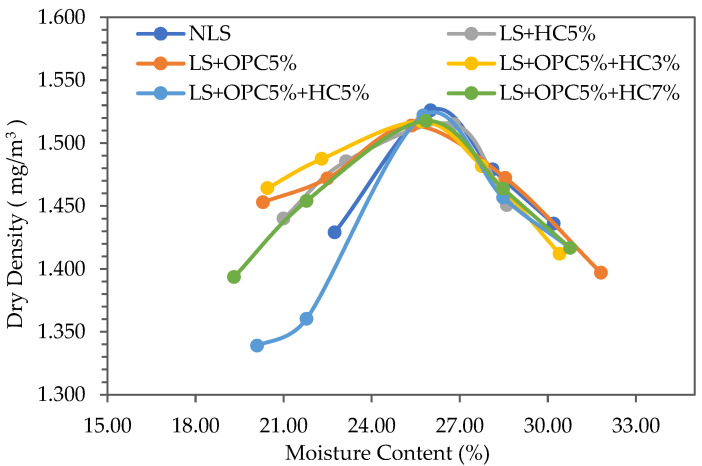
Result of the compaction test on the lateritic soil and cement mixed with caltite.

**Figure 2 materials-15-03087-f002:**
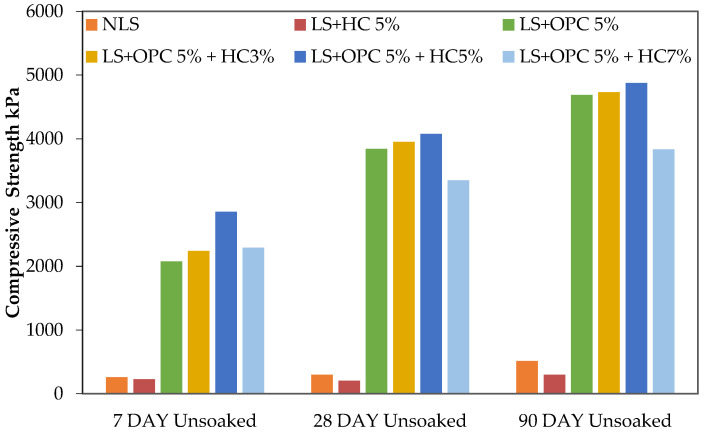
Compressive strength for curing in air (7, 28, and 90 days unsoaked).

**Figure 3 materials-15-03087-f003:**
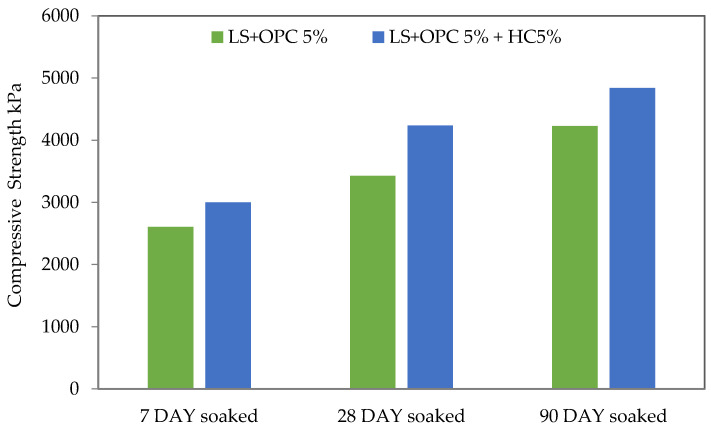
Compressive strength for curing in air (7, 28, and 90 days soaked).

**Figure 4 materials-15-03087-f004:**
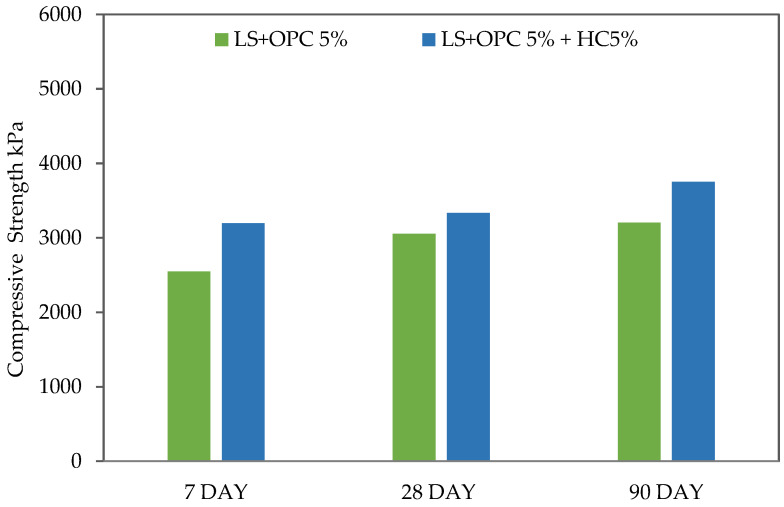
Compressive strength for curing ages of 7, 28, and 90 days under water.

**Figure 5 materials-15-03087-f005:**
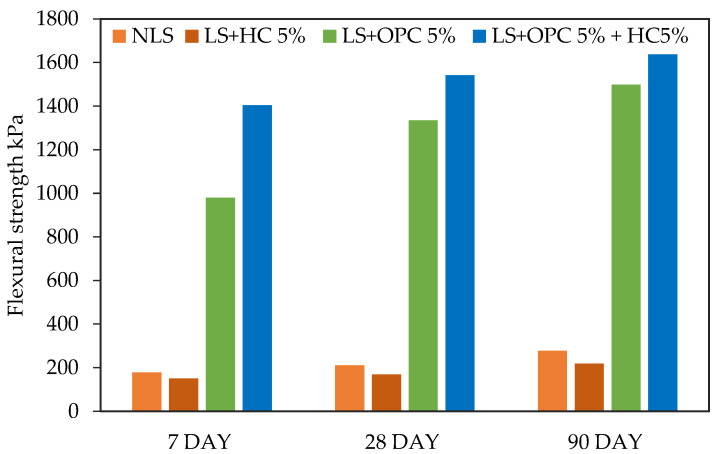
Flexural load–deflection curves of the treated and untreated test samples after 7, 28, and 90 days of curing.

**Figure 6 materials-15-03087-f006:**
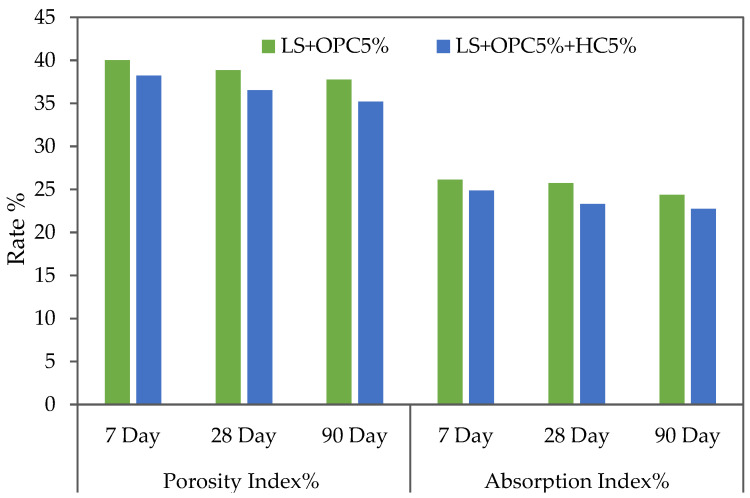
Rate of porosity for cemented lateritic soil with and without caltite.

**Figure 7 materials-15-03087-f007:**
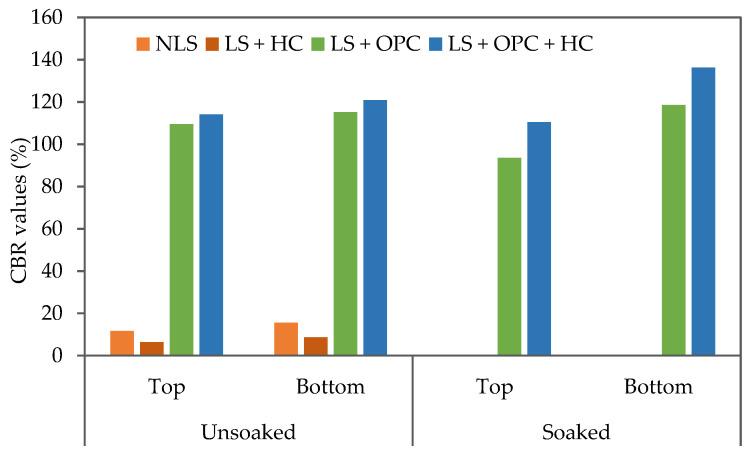
CBR of caltite–cement materials at a 7-day curing period with and without soaking under water.

**Figure 8 materials-15-03087-f008:**
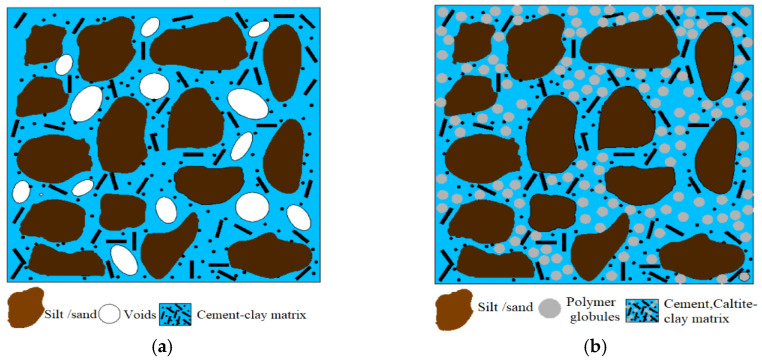
Schematic of the mechanism for strength components by (**a**) cement and (**b**) cement–caltite by polymer globules gathering in the cracks and void cement soil, reinforced by caltite.

**Figure 9 materials-15-03087-f009:**
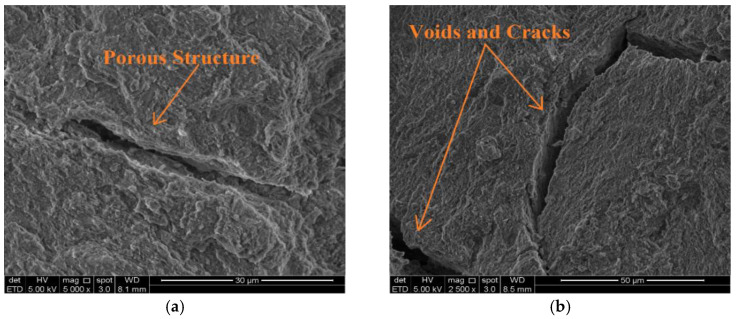
(**a**,**b**) SEM images of the soil treated by cement after 28 days of curing period.

**Figure 10 materials-15-03087-f010:**
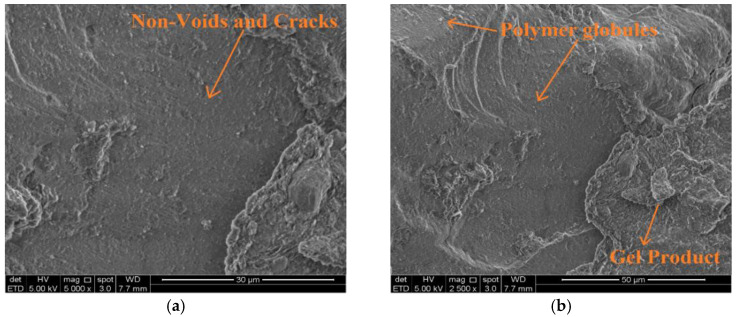
(**a**,**b**) SEM images of the soil treated by caltite with cement after 28 days of curing period.

**Figure 11 materials-15-03087-f011:**
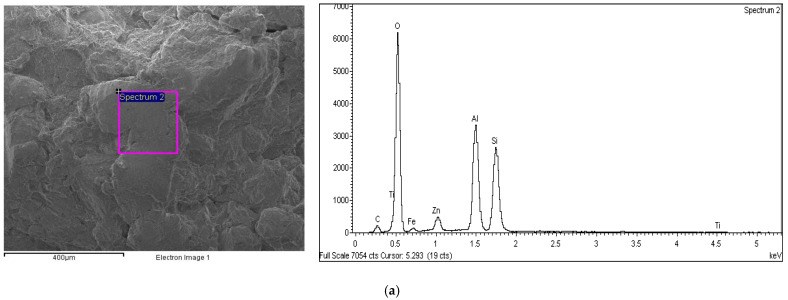
Twenty-eight-day curing period of the EDX spectrum for: (**a**) NLS, (**b**) LS + 5% OPC, and (**c**) LS + OPC + 5% HC.

**Table 1 materials-15-03087-t001:** Physical properties of lateritic soil and caltite used in this study. All units in the soil properties are in % except *.

Soil Properties	Quantity (%)
Specific gravity	2.6
Sand	41.6
Silt	42.1
Clay	16.4
* USCS classification	Silty sand (SM)
Liquid limit (LL)	68
Plastic limit (PL)	45.7
Plasticity index (PI)	22.3
Shrinkage limit (SL)	10.8
Optimal moisture	24
* Maximal dry density (mg/m^3^)	1.5
pH	4.8
Caltite properties	
Phase	Liquid
Specific gravity	0.98
Density (g/cm^3^)	1.00
Viscosity (m Pa s)	1.5
Color	Brown
pH	10.2
Cement (OPC) properties	
SiO_2_	21.8
Al_2_O_3_	4.7
Fe_2_O_3_	3.2
CaO	64.9
MgO	0.81
P_2_O_5_	0.08
K_2_O	1.2
Na_2_O	0.09
SO_3_	3.7
TiO_2_	0.2

**Table 2 materials-15-03087-t002:** Designation mix used in this study.

Types of Soil	Mixture Design	Curing Period Days	Repetitive Sample	Test Name	Number of Samples
NLS	Natural lateritic soil	7, 28, and 90	3	UCS–FS	18
LS + OPC 5%	Lateritic soil with cement 5%	7, 28, and 90	3	UCS–FS–PA	45
LS + HC 5%	Lateritic soil with HC 5%	7, 28, and 90	3	UCS–FS–PA	45
LS + OPC 5% + HC 3%, 5%, 7%, and 10%	Lateritic soil with cement 5% and HC 3%, 5%, 7%, and 10%	7, 28, and 90	3	UCS–FS–PA	63
Total No. of Sample					171

NLS = natural lateritic soil; LS = lateritic soil; OPC = ordinary Portland cement; HC = hydrophobic caltite; UCS = unconfined compressive strength; FS = flexural strength; PA = porosity and absorption.

**Table 3 materials-15-03087-t003:** Summary of the laboratory CBR values.

Sample No.	Soil NLS%	Caltite HC %	Cement OPC %	Curing Time Day	CBR %				
					Unsoaked		Soaked		Average %
					Top	Bottom	Top	Bottom	
NLS	100	0	0	7	11.62	15.56	-	-	13.59
LS + HC	100	5	0	7	6.41	8.69	-	-	7.55
LS + OPC	95	0	5	7 + 4	109.5	115.2	93.6	118.5	112.3–106
LS + OPC + HC	95	5	5	7 + 4	114.1	120.8	110.4	136.2	117.4–123.3

**Table 4 materials-15-03087-t004:** EDX analysis for untreated and treated lateritic soil after the compressive strength test at a curing period of 28 days.

Mixture	Curing Period (Days)	Element (wt%)								
		Silicon (Si)	Aluminum (Al)	Calcium (Ca)	Iron (Fe)	Carbon (C)	Titanium (Ti)	Oxygen (O)	Ca/Si Si/Ca	Si/Al
NLS	28	33.02	29.80	–	3.15	27.25	0.66	53.60	–	1.10
LS + OPC 5%	28	40.10	33.40	4.74	10.65	11.11	–	62.83	0.11 8.45	1.20
LS + OPC 5% + HC 5%	28	38.83	24.57	4.47	3.33	28.80	–	64.38	0.12 8.68	1.58

## Data Availability

Not applicable.

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
