# Peer review of "Hydrophobic Effect of Soil Stabilization for a Sustainable Subgrade Soil Improvement"

_materials, 2022, doi:10.3390/ma15093087_

Round 1

Reviewer 1 Report

This study investigated the chemical use of hydrophobic caltite and cement to enhance laterite soils using a range of tests. The present topic falls within the scope of the journal. I praise the good work done by the authors. However, I cannot propose publication in its current form because of some professional issues. The following comments and/or suggestions would help the authors to make this paper more readable and professional.

General comments:

  1. Abstract: Why not any quantitative results? Please provide some quantitative results here. Further, this part should present in the following order: background, methodology, material, results, and findings after research.

  1. Introduction: The research objectives are recommended to present at the end of this section. Further, the state-of-the-art research should be cited and discussed towards highlighting the significance of this study. To benefit our readers and provide further narrative, researches of phyco-mechanical properties of soil exposed to chemicals should also be included in the ‘Introduction’ section.

To this end, I would like to bring your attention to the following articles for your inclusion in the background to reinforce / echo the usefulness of your manuscript: A comparative study of different machine learning algorithms in predicting EPB shield behavior: a case study at Xi’an metro, China, 2021. Acta Geotechnica, 16(12): 4061-4080; Improvement of the Shearing Behaviour of Loess Using Recycled Straw Fiber Reinforcement, 2021. KSCE Journal of Civil Engineering, 25(9), 3319-3335.

  1. L86: The authors should define clearly the abbreviations before their first use (e.g. FS and PA).

  1. L95-95: The authors mentioned ‘Excavations of lateritc soil were made one metre below the ground's surface in Permatang Kerat Telunjuk, Bandar Baharu, Kedah.’. Is the investigation of water on the lateritic soil at this depth important? What are the implications?

  1. Table 1: The data should have the decimal members properly. For example, optimal moisture, liquid and plastic limits should have one decimal.

  1. L199-201: The authors mentioned ‘This pattern represented a value around eleven times higher than the untreated soil strength and half twice as great as the cement-treated soil strength’. The combination of treated samples with 5% caltite-cement (LS + OPC + HC) reached a compressive strength of 2853 kPa after a 7-day curing period, compared to the addition of 5% cement alone (2078 kPa). How do you say this pattern represented a value around half twice the cement-treated soil strength?

  1. Fig.10: In order to highlight the microscopic changes of the treated lateritic soil, it is suggested that the authors mark the quantitative data in the map and give further explanations.

Detailed comments:

  1. L145-150: The authors provide two methods: What is the difference between the two methods? What are the implications?

  1. L169-170: The authors mentioned ‘The dry density reduction could be explained by particle flocculation and accumulation caused by the rapid exchange of cations in the soil-stabiliser mixture’. How do the particle flocculation and accumulation caused by the rapid exchange of cations in the soil-stabiliser mixture affect the dry density reduction? Further elaboration should be supplemented in the revised manuscript.

  1. L190-193: The authors mentioned ‘there was a smaller rise in UCS when, 3% of caltite were added. In contrast, when 5% caltite was added, the UCS increased. In general, the differences in compressive strength observed in the soil with the 5% cement mixture and 5% caltite addition appeared to be the greatest’. When 7% calcite was added, the UCS decreased. No explanation was provided.

  1. L348-350: The authors mentioned ‘However, the cement-stabilised soil micrograph showed the presence of voids, porosity, and cracking, which primarily contribute to the loss of strength when underwater’. The authors are strongly advised to strengthen their explanations.

Author Response

The authors would like to express their appreciation to the reviewer for his effort and time given in reviewing the article and for providing valuable comments and suggestions, and all the comments have been revised.

Reviewer 2 Report

The manuscript presents an experimental study to investigate effects of using caltite on cement-stabilized soil. What’s described in lines 182-185 do not seem to match what’s presented in Figure 2 and Figure 3. Captions of Figure 3 and Figure 4 could be misleading and suggest the same sample group. If samples in Figure 2 and Figure 3 are cured in air, please state so in the caption. Please explain how the reduction (Line 221) in UCS is observed on day 28 and day 90. Figure 4 shows there was a strength gain on day 90 although the gain was small.

As the study also attempts to investigate the effects of caltite addition on the durability of stabilized soil, the study should conduct a wet-dry cyclic test.

There are numerous grammatic errors and typos. English of the manuscript needs to be carefully edited and proof read. Some editorial comments are listed below:

Line 11: please change “still questionable” to “is still questionable”

Line 14: please change “evaluation of soil” to “evaluation of soil properties”

Line 30: please change “degradation in water” to “resistance to degradation in water”

Line 140: please change “discarded” to “scraped”

Line 148: please change “im-mersed” to “immersed”

Figure 1 and Figure 2: the natural lateritic soil is designated as NLS or LS. They are not consistent in those two figures.

Line 149-150: this is not clear. Please rewrite.

Line 200: please make correction to the sentence

Line 213: please change “an increased” to “an increased strength”

Line 236 and 237: please rewrite. It’s not clear.

Author Response

(The authors gave the same response as above.)

Round 2

Reviewer 1 Report

I don't have further comments.

Reviewer 2 Report

The reviewer appreciates the authors' efforts expended in improving the manuscript. The authors have adequately addressed the reviewer's comments and would like to recommend acceptance for publication.